# Dietary Effects of Introducing Salt-Reduced Bread with and without Dietary Counselling—A Cluster Randomized Controlled Trial

**DOI:** 10.3390/nu14183852

**Published:** 2022-09-17

**Authors:** Nanna Louise Riis, Anne Dahl Lassen, Kirsten Bjoernsbo, Ulla Toft, Ellen Trolle

**Affiliations:** 1National Food Institute, Technical University of Denmark, 2800 Kgs. Lyngby, Denmark; 2Centre for Clinical Research and Prevention, Bispebjerg and Frederiksberg Hospital, 2000 Frederiksberg, Denmark

**Keywords:** salt reduction, sodium, potassium, RCT, behavior change, dietary intervention, public health strategies, food policies, reformulated processed foods, cardiovascular disease prevention

## Abstract

Successful strategies for policy makers and the food industry are required to reduce population salt intake. A 4-month cluster randomized controlled trial was conducted to evaluate whether the provision of salt-reduced bread with or without dietary counselling affected the dietary intake of selected food groups, energy, macronutrients, sodium, and potassium. Eighty-nine families (*n* = 309) consisting of minimum one parent and one child were assigned to receive bread gradually reduced in salt content alone (Intervention A), combined with dietary counselling (Intervention B), or bread with regular salt content (control). Food intake was recorded for seven consecutive days at baseline and follow-up. Salt intake was reduced in both Intervention A (−1.0 g salt/10 MJ, *p* = 0.027) and Intervention B (−1.0 g salt/10 MJ, *p* = 0.026) compared to the control. Consumption of bread and both total and salt-rich bread fillings remained similar between groups, while ‘cheese and cheese products’ were reduced in Intervention A (−38%, *p* = 0.011). Energy intake and macronutrient distribution were not affected in Intervention A, but Intervention B resulted in a higher energy intake (512 kJ, *p* = 0.019) and a lower energy % (E%) from saturated fat (−1.0 E%, *p* = 0.031) compared to the control. In conclusion, provision of salt-reduced bread both with and without dietary counselling successfully reduced dietary salt intake without adversely affecting the dietary nutritional quality.

## 1. Introduction

High levels of dietary salt have repeatedly been associated with raised blood pressure (BP), a leading risk factor for cardiovascular disease (CVD) [1,2,3,4]. Increasing evidence has also found associations between high salt consumption and increased risk of stomach cancer, renal disease, and osteoporosis [5,6,7,8,9]. High salt consumption has been estimated to account for three million deaths annually, thereby being the dietary risk factor accounting for most deaths globally [10]. In most countries, salt intake far exceeds the recommended level of less than 5–6 g/day [11,12,13,14], and in Denmark, the average adult consumption is estimated to be 9.5 g/day [15].

Processed food is the main source of dietary salt in western countries, including Denmark, and accounts for approximately 70% of total salt consumption [16,17]. Achieving substantial and sustainable reductions in population salt intake therefore requires decreased levels of salt in the available food products, and asimultaneous education and encouragement of consumers to purchase lower-salt food products [13]. Gressier et al. found that the sodium density of foods and beverages consumed by the UK population decreased by 17% between 2008/09 and 2016/17. The authors concluded that this decrease was largely driven by reformulated food products [18]. Food products with the greatest contribution to dietary salt include bread and bakery products, cheeses, spreads, and processed meat and fish products used for bread fillings, etc. [19]. In Denmark, bread alone contributes to approximately 20% of total salt consumption [15], making salt reduction in bread a potentially effective strategy to reduce population salt intake. An important reason for adding salt to food during manufacturing is to increase palatability to satisfy consumer preferences [20,21]. Abrupt and large reductions in salt content may reduce consumer acceptance [20], but by reducing the salt content gradually, consumers have been found to become accustomed to the changed taste, whereby the new level of salt content remains unnoticed by most consumers [22,23,24]. Promisingly, previous research has found reductions in salt content between 25% and 50% in bread to be well accepted by consumers [23,25], with bread consumption remaining unchanged by this reduction [25]. However, bread is typically consumed with different types of spreads and fillings, and the reduced salt flavor may be compensated for by choosing more salt-rich bread toppings. For example, in a consumer study, overall saltiness perception and pleasantness increased when butter was added to salt-reduced bread [26]. If the consumption of salt-reduced bread results in compensating behaviors, the reformulation will not lead to the targeted reduction in dietary salt intake. Additionally, the consumption of salt-reduced food may unintentionally influence other aspects of the diet. For instance, a randomized controlled trial (RCT) found that substantial reductions in salt content of 67% in bread caused a decline in bread consumption [25]. In contrast, another experimental study found that ad libitum consumption of soup low in salt was higher than ad libitum consumption of soup high in salt [27].

Although large proportions of salt are consumed from processed foods, the reformulation of food products to contain less salt may need to be combined with other salt reduction strategies, to reach the recommended level of salt intake. The use of dietary counselling and other behavior change interventions in adults, including face-to-face and online interventions, have shown promising results in previous intervention studies [28,29,30,31]. A systematic review conducted by Khalesi et al. concluded that behavior change interventions can reduce salt purchasing and salt use during cooking, improve label reading, enhance the selection of lower-salt options, and drive the use of salt substitutes [31]. For example, in an RCT, Australian consumers were educated on how to identify food products with lower salt content either by (1) using a label on the front of the pack stating that the product adhered to nutritional criteria for salt reduction or (2) by using the information of the nutrition declaration typically on the back of the pack. The daily salt intake was reduced by 0.9 g/day by using the first type of label and 1.9 g/day by using the second type of label [28]. Another RCT, conducted in the US, investigated the effectiveness of educating consumers to use herbs and spices to replace salt during cooking, which resulted in a daily salt reduction of 1.0 g/day [29].

Research investigating the dietary effects, including potential compensating behaviors, when introducing salt reduction interventions in a real-life context is lacking. Thus, the objective of the present study was to estimate the effect of providing salt-reduced bread both with and without dietary counselling on the dietary behavior, i.e., intake of selected food groups, energy intake, and nutrients, among Danish families.

## 2. Materials and Methods

### 2.1. Study Design and Population

The SalT Reduction InterVEntion (STRIVE) study was a cluster RCT with a parallel design. Danish families were recruited from five municipalities in the southwestern part of the Capital Region of Denmark through social media at schools, kindergartens, and larger companies, word of mouth, and posters in the local area. Families were randomly allocated, using a computer-generated sequence of random group assignment, into one of three groups, receiving bread gradually reduced in salt content (Intervention A), bread gradually reduced in salt content combined with dietary counselling (Intervention B), or bread with a regular salt content (control) over a period of four months. Families were unaware of whether they received regular or salt-reduced bread. A thorough description of the methodology has been provided elsewhere [32]. To be included in the study, families had to consist of at least one parent (18–65 y) and one child (3–17 y). Average bread consumption among adults should be around 175 g/day or more, which was expected to be fulfilled, if participants reported daily consumption of bread. Exclusion criteria included self-reported use of cholesterol-lowering or antihypertensive medication, pregnancy, diabetes, coronary heart disease, celiac disease, gluten intolerance, and urine albumin > 300 mg/day calculated from a spot urine test at the baseline health examination.

### 2.2. Intervention

During the intervention, families received bread either gradually reduced in salt content or with regular salt content twice a week (Mondays and Thursdays). The bread was provided free of charge and included a mixture of rye bread and wheat bread in the form of a loaf or as buns. The amount was adjusted to fit the families’ habitual bread intake. An overview of the salt content in the studied bread products can be found in Table 1. During the first two weeks of intervention, all families received bread with a salt content of 1.2 g/100 g, corresponding to the average content in Danish supermarket and bakery bread [33]. This level of salt content was maintained during the rest of the intervention in the control group. In the two intervention groups, the salt content in bread was reduced by 0.2 g/100 g every week until a level of 0.6 g/100 g and 0.4 g/100 g was reached in rye bread and wheat bread, respectively, which was then maintained during the remainder of the intervention (difference in salt content of 50% and 67% for rye bread and wheat bread, respectively). Rye bread was not reduced to the same extent as wheat bread due to undesirable effects on the texture. In order to minimize the risk of decreasing participants’ acceptance of project bread, when reducing salt content considerably, a liquid low-sodium salt [34] was added to the salt-reduced bread to slightly increase the salty taste. Thus, bread with a salt content of 0.4 g/100 g was aimed to taste equivalent to a salt content of 0.55 g/100 g.

In combination with salt-reduced bread, dietary counselling was provided in Intervention B. This consisted of one group counselling (including a general presentation and three workshops), one individual family counselling with two follow-up telephone calls, and weekly e-mails. The counselling sessions included five main messages: (1) buy less salt-rich foods within different food categories by checking nutrition declaration and/or selecting products with the keyhole label [35]; (2) eat less food with a high salt content, such as processed meats, cheese, and convenience food; (3) reduce the use of salt during cooking and at the dining table; (4) flavor food with herbs and spices as an alternative to salt; (5) follow the Danish plate model, where 40% of the plate is filled with fruit and vegetables.

### 2.3. Outcome Measures

#### 2.3.1. Background Characteristics

At baseline and follow-up, all families went through a health examination at the Center for Clinical Research and Prevention, Glostrup, Denmark. Anthropometric measurements including height and body weight were measured without shoes and in light clothing using a Harpenden stadiometer and a digital scale, respectively. Body Mass Index (BMI) was calculated as weight (kg)/height^2^ (m^2^). Information regarding participants’ physical activity level, smoking status, and parental education was obtained from questionnaires. Physical activity level was divided into sedentary (inactive), moderate (moderately active > 4 h/week), and vigorous activity (strenuous activity > 3 times/week). Participants were categorized as smokers if they smoked daily or occasionally and non-smokers if they never smoked or smoked once but stopped. Parental education was divided into (1) minimum secondary school completion, (2) bachelor’s degree or equivalent (3–4 y), and (3) post-graduate degree (≥4 y), and all family members were assigned according to the parent in the family with the highest education.

#### 2.3.2. Dietary Assessment

All family members were asked to complete seven consecutive days of food recording at baseline and follow-up. In connection to the health examinations, trained research staff instructed families on how to use the dietary assessment tool, and parents were instructed on how to support and/or complete the registration for younger children. Daily intake of food and beverages was recorded in a validated web-based dietary assessment software program [36,37] that was slightly modified to fit the purpose of the study. Pre-coded dishes were updated to better distinguish between composite food products containing either high- or low-salt ingredients, and the food composition database was updated with respect to sodium content. The project bread was included in the dietary assessment software at follow-up. The recordings were entered during or at the end of the day and followed a typical Danish meal pattern including breakfast, a morning snack, lunch, an afternoon snack, dinner, and an evening snack. Participants could choose from a variety of pre-coded food products and beverages, as well as typical Danish dishes. An open answer option was also available if none of the pre-coded response categories matched the consumed product. The portion size was estimated using photos with varying amounts, and participants could then specify the number of servings. For food recorded in the open answer option, the portion size was written in grams or as household measures. After each main meal, built-in prompts appeared to remind participants to record beverages, cakes, and sweets.

During the recording periods, a telephone hotline was open to help families who experienced problems. If recordings were not completed, reminder e-mails were sent to the families, and when two recordings were missing, a telephone call was made to find out if the family had encountered any problems.

#### 2.3.3. Estimation of Dietary Intake

For each participant, the average dietary intake per day was estimated based on the intake of the days recorded, which was at least four days for each participant. To calculate the intake of energy, nutrients, and foods, a General Intake Estimation System (GIES) (originally developed for DANSDA [15]) was used to interpret data entered in the food records into ingredients. The ingredients were linked to nutrient data from a custom version of the Danish Food Composition Databank version 3 [38].

#### 2.3.4. Selection of Food Groups

Selected food products were assigned to relevant food groups and the inclusion of food groups was based on the aims to investigate the interventions’ effects on: (1) bread consumption, as it may be affected by reducing the salt content in bread and/or by providing bread free of charge; (2) consumption of ‘butter and spreads’ and choice of bread fillings, as salt-containing bread fillings or cold cuts may compensate for the reduced salt flavor in bread; (3) intake of high-salt food groups, including ‘cheese and cheese products’ and ‘processed red meat, poultry, and fish’, as it may compensate for a lower salt intake from bread-containing meals; (4) intake of food groups that may be substituted with bread when receiving this free of charge, such as other starchy products including ‘rice and pasta’, ‘potato and potato products’, and ‘breakfast cereals’; (5) intake of other main food groups within the diet that might have an impact on health and/or and nutrient intake, including ‘milk and milk products’, ‘fruit’, ‘vegetables’, ’red meat and poultry’, ’fish and fish products’, ‘cakes, sweets, and chocolate’, and ‘sugar-sweetened beverages’.

Bread fillings were assigned to high- (>2.5 g/100 g), medium- (1–2.5 g/100 g), and low- (<1.0 g/100 g) salt bread fillings based on the generic salt content obtained from the Danish Food Composition Databank version 3. The main food products assigned to high-salt bread fillings included sliced red meats and poultry, smoked and marinated fish, and some types of cheeses. Medium-salt fillings included sliced red meat and poultry products, canned or mildly processed fish, and firm cheeses. Low-salt fillings included fruit and vegetables, eggs, canned fish, soft cheeses, and sweet fillings such as Nutella and thin slices of chocolate.

#### 2.3.5. Under-, Acceptable, and Over-Reported Intake of Energy

Participants were classified as under-reporters (UR), acceptable reporters (AR), or over-reporters (OR) of energy intake based on the ratio between the mean reported energy intake and the basal metabolic rate (BMR) using the Goldberg cut-off values at the individual level, as suggested by Black [39,40]. BMR was estimated from weight and height using age- and gender-specific Scofield equations [41]. Physical activity level was based on answers in the questionnaire and set to 1.6, 1.8, and 2.0 in children and 1.4, 1.6, and 1.8 in adults for sedentary, moderate, and vigorous physical activity, respectively [42].

### 2.4. Statistical Analysis

All analyses were conducted using R version 3.5.3 (R Core Team, 2019, Vienna, Austria). The outcomes are presented in g/10 MJ to account for differences in intake between children and adults. For macronutrients, the outcomes are presented in Energy % (E%). All analyses have also been conducted separately for children and adults, and these results are presented in the Appendix A.

Categorical data are presented as frequencies and percentages and continuous data are presented as mean and standard deviation (SD) or median and interquartile range (IQR) according to distribution. Distributions were tested using histograms, and all selected food groups were log-transformed before analysis to meet assumptions for normal distribution. Participants without an intake of a food group were included with zero values in the presentation of intakes at baseline, and a value of 0.01 g/day was inserted before log transformation to include these participants in the analyses. The intention-to-treat approach was used in all analyses, and missing values were imputed with 100 samples. Compound symmetry was chosen as the variance structure. To estimate baseline differences between groups, mixed models were used, with baseline value as the outcome variable, treatment group as a fixed effect, and family as a random effect.

Changes from baseline to follow-up within groups were estimated using mixed models with the baseline and follow-up values as outcome variables; baseline/follow-up visit, age, gender, BMI, parental education, and under- and acceptable reported energy intake as fixed effects, and participant as a random effect. Family was not included as a random effect as this caused variance estimates to approach zero.

Differences between groups were analyzed using mixed models, with follow-up value as the outcome variable; treatment group, baseline value, age, gender, BMI, parental education, and under- and acceptable reported energy intake as fixed effects, and family as a random effect. The intraclass correlation (ICC) is reported to describe the resemblance of values within the clusters (families). When analyzing the food group ‘butter and spreads’, family was not included as a random effect due to variance estimates approaching zero. Participants with an over-reported energy intake only included one participant at baseline and two participants at follow up; thus, these participants were assigned as acceptable reporters of energy intake, due to imputation difficulties when having few cases.

## 3. Results

In total, 89 families (individual family members *n* = 309) were included in the study. Twenty-five families (*n* = 81) were assigned to Intervention A, 35 families (*n* = 127) to Intervention B, and 29 families (*n* = 101) to the control group. By chance, more families were randomized to Intervention B, which is why the number of participants is also higher in this group. During the study, seven participants in Intervention A, seven participants in Intervention B, and six participants in the control group dropped out (*n* = 20). The reasons for dropout included lack of time (*n* = 10), not showing up for follow-up assessments (*n* = 4), the taste of the bread (different from the usual bread) (*n* = 2), family issues (*n* = 3), and moving to another city (*n* = 1). One participant had missing dietary data at follow-up, six participants had missing data on BMI, and eleven participants had missing data on parental education, physical activity, and smoking status. It was not possible to assess whether energy intake was under-, acceptable, or over-reported for 16 participants at baseline and 23 at follow-up (including dropouts), due to missing dietary registrations or missing values on weight, height, or level of physical activity. At baseline, no significant differences between groups were seen for gender, age, BMI, physical activity, or smoking status (Table 2).

### 3.1. Food Groups

Changes in the intake of food groups from baseline to follow-up in the three groups are presented in Table 3. The intake of ‘bread, total’ increased significantly from baseline to follow-up for all three groups, with 27% (*p* = 0.000), 22% (*p* = 0.000), and 16% (*p* = 0.002) in Intervention A, Intervention B, and the control group, respectively. At baseline, all three groups consumed more ‘wheat bread’ than ‘rye bread’. During the intervention, it was primarily the intake of ‘wheat bread’ that increased in Intervention A (37%, *p* = 0.001) and in the control group (18%, *p* = 0.117), whereas the intake of both ‘wheat bread’ and ‘rye bread’ increased in Intervention B (27%, *p* = 0.002 and 41%, *p* = 0.036, respectively). At follow-up, project bread accounted for an average of 80%, 85%, and 88% of the total bread consumption in Intervention A, Intervention B, and in the control group, respectively. Consumption of ‘butter and spreads’ did not change in any of the three groups during the study, whereas the intake of ‘bread fillings, total’ was significantly reduced in Intervention B (−17%, *p* = 0.017) and borderline significantly reduced in the control group (−15%, *p* = 0.051). The reduced intake of ‘bread fillings, total’ in the control group was caused by a significant decrease in both ‘medium-salt fillings’ and ‘low-salt fillings’ (−31%, *p* = 0.024 and −43%, *p* = 0.004, respectively) but not in ‘high-salt fillings’, while the reduced intake of ‘bread fillings, total’ in Intervention B was due to a non-significant decrease in all three groups of bread fillings (high-, medium-, and low-salt fillings).

The intake of ‘cheese and cheese products’ was significantly reduced in both Intervention A (−35%, *p* = 0.005) and Intervention B (−17%, *p* = 0.044) during the study, and in Intervention A, there was also a simultaneous increase in the intake of ‘processed red meat, poultry, and fish’ (26%, *p* = 0.024). The intake for the high-salt food groups remained unchanged in the control group.

The intake of ‘breakfast cereals’ was reduced in the control group (−52%, *p* = 0.006), whereas no other significant changes in starchy products such as ‘rice and pasta’ and ‘potato and potato products’ were seen in any of the groups. The intake of some of the other food groups in the diet that may have a general health impact changed during the study period. The intake of ‘milk and milk products’ was significantly reduced in both Intervention A (−12%, *p* = 0.030) and Intervention B (−15%, *p* = 0.007) during the study, whereas the intake of ‘fruit’ was significantly increased in these groups (46%, *p* = 0.009 and 56%, *p* = 0.000, respectively). Consumption of ‘red meat and poultry’ increased significantly in Intervention A (19%, *p* = 0.002), while the intake of ‘fish and fish products’ decreased in both Intervention A and the control group (−61%, *p* = 0.003 and −52%, *p* = 0.004, respectively). The intake of ‘cakes, sweets, and chocolate’ and ‘sugar-sweetened beverages’ remained unchanged, apart from Intervention B, where the intake of ‘cakes, sweets, and chocolate’ was decreased by −25%, *p* = 0.017, from baseline to follow-up.

Differences between groups at follow-up (adjusted for baseline and other covariates) are presented in Table 4. The intervention did not result in any group differences in the consumption of ‘bread, total’.

Consumption of ‘butter and spreads’ and ‘bread fillings, total’, as well as bread fillings with varying salt content, did not differ between groups.

The intake of ‘cheese and cheese products’ was significantly lower in Intervention A compared to the control group (−38%, *p* = 0.011). The intake of starchy food groups did not differ between groups at follow-up, except for ‘breakfast cereals’, which was significantly higher in Intervention B compared to the control group (130%, *p* = 0.033).

There was a borderline significantly higher intake of ‘fruit’ in Intervention B compared to the control group (31%, *p* = 0.066). The intake of ‘fish and fish products’ was significantly higher in Intervention B compared to Intervention A (171%, *p* = 0.012), while the intake of ‘red meat and poultry’ was borderline significantly lower (−18%, *p* = 0.057). No differences between groups were found for the consumption of ‘cakes, sweets, and chocolate’ and ‘sugar-sweetened beverages’.

### 3.2. Nutrients

Changes from baseline to follow-up in the intake of energy and nutrients are presented in Table 5. Sodium consumption decreased in Intervention A (−0.4 g/10 MJ (−1.0 g salt/10 MJ), *p* = 0.000) and Intervention B (−0.5 g/10 MJ (−1.3 g salt/10 MJ), *p* = 0.000), while no changes were found in the control group. Dietary potassium remained unchanged in all three groups.

Energy intake was borderline reduced in Intervention A (−406 kJ, *p* = 0.061) and significantly reduced in the control group (−394 kJ, *p* = 0.008), whereas the energy intake was increased in Intervention B during the study (325 kJ, *p* = 0.065). Based on observed data, the proportion of participants with an underreported energy intake at baseline was 30% in Intervention A, 28% in Intervention B, and 24% in the control group. At follow-up, this was 30%, 35%, and 32%, respectively. The macronutrient distribution mainly remained unchanged during the study, but in Intervention A, there was a significant increase in E% from fat (1.0 E%, *p* = 0.037), and in Intervention B, there was a significant reduction in E% from saturated fat (−0.8 E%, *p* = 0.001), whereas the contribution of energy from added sugar increased (0.8 E%, *p* = 0.045). In the control group, there was a significantly reduced intake of dietary fiber (−2 g/10 MJ, *p* = 0.001).

Differences between groups at follow-up (adjusted for baseline and other covariates) are presented in Table 6. Dietary sodium was significantly reduced in Intervention A (−0.4 g/10 MJ (1.0 g salt/10 MJ), *p* = 0.027) and Intervention B (−0.4 g/10 MJ (1.0 g salt/10 MJ), *p* = 0.026) compared to the control group, whereas no differences between groups were seen in the intake of potassium.

The energy intake was significantly higher in Intervention B compared to both the control group (512 kJ, *p* = 0.019) and Intervention A (554 kJ, *p* = 0.016). There were no differences between groups in the macronutrient distribution, except for a lower E% from saturated fat in Intervention B compared to the control (−1.0 E%, *p* = 0.031) and Intervention A (−1.0 E%, *p* = 0.027). Analyses conducted separately for children and adults showed that the intervention had similar effects on both children and adults (Appendix A).

## 4. Discussion

This RCT providing salt-reduced bread to families with and without dietary counselling showed that salt intake, estimated from dietary records, decreased both when providing salt-reduced bread alone and in combination with dietary counselling. The consumption of bread increased in all three groups but did not differ between the intervention groups receiving salt-reduced bread and the control group receiving bread with a regular salt content. No compensating behavior in terms of the choice of bread fillings or other salt-rich food groups was identified. There was a tendency towards a higher intake of fruit when receiving salt-reduced bread in combination with dietary counselling, and this intervention also resulted in a higher intake of breakfast cereals compared to the control. The energy intake and macronutrient distribution were not affected by receiving salt-reduced bread alone, but the addition of dietary counselling resulted in a higher energy intake and a lower E% from saturated fat.

Previous studies have investigated the dietary impact of providing salt-reduced food products [25,43]. In an RCT by Bolhuis et al., participants were provided with bread with a regular salt content (1.8 g/100 g) or bread with a reduced salt content of 52% (0.87 g/100 g) or 67% (0.6 g/100 g) [25]. The bread consumption remained unchanged when the salt content in bread was reduced by 52%, but when the salt content was reduced by 67%, the bread consumption decreased. This contrasts with the findings in the present study, where the consumption of bread with a reduced salt content of 67% did not differ from that of bread with regular salt content (1.2 g/100 g) provided to the control group. The study by Bolhuis et al. only lasted for four weeks and the bread products were only provided for breakfast on weekdays. Thus, the decreased bread intake could result from an insufficient exposure period, as a preference for food with a reduced salt content may develop gradually over time, with regular consumption [44]. Notably, the present study provides evidence that large reductions in the salt content of bread (67%) over a long period of time (4 months) do not reduce bread consumption. This may in part result from an increased salt taste sensitivity and higher liking of bread with a low salt content (0.4 g/100 g) when consuming salt-reduced bread, as observed in a previous publication based on this trial [45].

Another RCT, conducted by Janssen et al., investigated the effect of serving salt-reduced lunches for three weeks [43]. The salt content of the provided food products was reduced between 29% and 61% and included bread, cold cuts, cheeses, salad spreads, hot snacks, soups, and dressings. The intervention did not trigger salt-compensating behavior during the remainder of the day, and salt intake decreased by 2.5 g/day. Similarly, the present study showed no evidence of salt intake compensation by the intake of ‘high-salt fillings’, ‘butter and spreads’, or other high-salt food groups. In fact, the intake of ‘cheese and cheese products’ was reduced in Intervention A, however, this intervention also resulted in increased consumption of ‘processed red meat, poultry, and fish’ (non-significant). Nevertheless, this potential compensating behavior did not prevent a reduction in dietary salt. However, the reduction in dietary salt intake in the present study was modest (−1.0 g/10 MJ), and the greater reduction in dietary salt achieved in the study by Janssen et al. suggests that reformulation of several food products to contain less salt may be required to achieve substantial reductions in salt intake. Interestingly, the study by Janssen and colleagues took place in a research canteen, and this study demonstrates that salt intake can also be reduced in a real-life context by providing reformulated food.

Regarding the intake of other food groups, it was only the intake of ‘breakfast cereals’ that was significantly affected, with a reported difference in intake at follow-up of 130% in Intervention B compared to the control group. Although the intake of ‘breakfast cereals’ appeared to be affected to a great extent, the intake was low at baseline (27 g/10 MJ) and decreased in both Intervention B and the control group from baseline to follow-up. Thus, the absolute consumption did not seem to differ considerably between groups. Promisingly, the intake of other food groups that may have an impact on health, including the sugar-containing food groups ‘cakes, sweets, and chocolate’ and ‘sugar-sweetened beverages’, remained similar between groups. It was only the intake of fruit that tended to differ between groups following the intervention, with a borderline significant increase in Intervention B, which was in accordance with the advice given in the dietary counselling to increase consumption of fruit and vegetables.

The reported energy intake increased among participants, especially adults, receiving salt-reduced bread with dietary counselling (Appendix A). Bolhuis and colleagues have demonstrated previously that the reformulation of food to contain less salt may result in higher consumption, as ad libitum intake of soup with a low salt content was greater than ad libitum intake of soup with a high salt content [27]. The provision of salt-reduced bread also resulted in higher consumption of bread at follow-up; however, this did not differ significantly from the consumption of bread with a regular salt content given to the control group. The increased bread consumption from baseline to follow-up may have resulted from the larger portion sizes of the intervention bread. The study bread products were added to the dietary registration tool at follow-up, and the portion sizes reflected the actual sizes of the study bread products. These sizes were larger than the generic bread products used for measuring bread intake at baseline and may thereby have resulted in a higher reported bread consumption at follow-up. This could also explain why the increased bread consumption did not result in a simultaneous increase in ‘butter and spreads’ and ‘bread fillings’.

The differences in the reported energy intake may result from differences in the quality of the dietary records, as findings from a previous publication revealed that BMI did not change in any of the three groups during the intervention [46]. It could be speculated that participants receiving dietary counselling increased their commitment to the intervention, and therefore provided a more detailed dietary registration at follow-up, resulting in a higher reported energy intake. The same level of commitment may not have been achieved among participants receiving project bread in the absence of dietary counselling (Intervention A and control group), and the quality of the dietary registrations may therefore have been reduced towards the end of the intervention, resulting in a reduction in the reported energy intake from baseline to follow-up. Moreover, it was primarily the fruit consumption that increased in Intervention B, and since a higher fruit intake was advised during the dietary counselling, it could be speculated that participants in Intervention B exaggerated their reported fruit intake to appear compliant or achieve approval from the nutrition counsellors. Nonetheless, the macronutrient distribution in both salt reduction intervention groups remained similar to the control group, except for a slightly lower E% from saturated fat among participants in Intervention B. Thus, with respect to the Nordic Nutrition Recommendations, there was no negative effect on the quality of the diet in terms of the macronutrient distribution. The reduction in dietary salt, combined with the finding that the dietary quality was not negatively affected, provides practical applications for policy makers and for the food industry to lower population salt intake, by increasing the production and availability of food products with a reduced salt content.

Some strengths and limitations of the study should be mentioned. The inclusion of healthy Danish families and the real-life context in which the study took place increase the generalizability of the results to the wider Danish population. Although it may not be realistic to provide dietary counselling to the whole population, the elements used during the dietary counselling sessions could provide a foundation for future public health education.

More families were by chance randomized to Intervention B. To ensure equal distribution of families in the three groups, block randomization should have been used. Nevertheless, the randomization was successful in ensuring that there were no significant differences between the three groups in the baseline characteristics.

The intake of foods and nutrients is presented in g/10 MJ to account for differences in energy intake between children and adults. It also accounts for changes in energy intake at follow-up, as the lower energy intake in Intervention A and the control group at follow-up might increase the absolute registered sodium reduction (g/day), while the higher energy intake in group B at follow-up might decrease the reduction in the absolute registered sodium intake.

The inclusion of dietary counselling in addition to salt-reduced bread did not result in a greater reduction in daily salt intake when the intake was estimated from dietary records. This contrasts with the findings from a previous publication based on this study, where the daily salt intake among adults, estimated from three 24-h urine collections, was lowered to a greater extent when dietary counselling was given in addition to salt-reduced bread [47]. Estimated salt intake based on repeated 24-h urine collections is regarded as the gold standard because this is an objective measure of dietary salt intake and because 93% of consumed sodium is excreted in the urine within 24 h [48]. The contrasting findings on salt intake may be due to the dietary registration tool not being sufficiently sensitive to correctly estimate changes in salt intake resulting from the advice given in the dietary counselling. Firstly, it was not possible for participants to register when they had substituted a food product with a low-salt version of the same product. Since an important piece of advice in the dietary counselling was to select food products with less salt, by checking the information of the nutrition declaration and/or selecting products with the keyhole label, this may have resulted in the reduction in salt intake being underestimated. Secondly, participants were advised to reduce their use of salt when cooking and at the dining table, but the use of discretionary salt was not measured in the dietary registration, further increasing the risk of underestimating reductions in salt consumption. Balancing the details required in food registration with the participation burden of the registration is always a challenge in food recording.

The power calculation was based on changes in dietary salt; hence, differences in the dietary intake of other nutrients and food groups may not have reached statistical significance due to a lack of power.

Lastly, the intake of bread at baseline (median intake between 153 and 160 g/10 MJ) was lower than the expected bread consumption of 175 g/day among adults, and this may partially explain why the effect on salt intake was modest. Moreover, consumption of bread supplied in the project was, in general, high, but not all the consumed bread products were accounted for by the project bread, and this may also have contributed to the modest reduction in salt intake. To achieve more substantial salt reductions, future studies should investigate the effect of providing several salt-reduced food products in a real-life context.

## 5. Conclusions

In conclusion, salt intake (sodium) was lowered by providing salt-reduced bread alone and in combination with dietary counselling. The amount of bread consumed increased in all three groups, but it did not differ between those receiving salt-reduced bread and those receiving bread with a regular salt content, indicating that the taste was accepted despite the reduced salt content. Furthermore, there was no evidence of compensating behavior by selecting more salt-rich bread fillings or other high-salt food groups; in fact, consumption of cheese and cheese products was lowered when receiving salt-reduced bread alone. The energy intake and the quality of the diet in terms of the macronutrient distribution and intake of food groups was not affected by providing salt-reduced bread alone, but adding dietary counselling may have increased energy intake and resulted in minor improvements in the dietary quality, as the E% from saturated fat was lowered and fruit consumption increased (borderline significant). Future studies should make use of improved and targeted food registration tools to better measure changes in intake. To achieve more pronounced reductions in dietary salt intake, future and larger studies should investigate the effect of providing several salt-reduced food products in a real-life setting. Combination with dietary counselling might increase the effect and promote healthier dietary intake in general.

## Figures and Tables

**Table 1 nutrients-14-03852-t001:** Salt content in study bread products.

Weeks from Intervention Start	Salt-Reduced Bread (g Salt/100 g)		Regular Salt Bread (g Salt/100 g)	
	Rye Bread	Wheat Bread	Rye Bread	Wheat Bread
1	1.2	1.2	1.2	1.2
2	1.2	1.2	1.2	1.2
3	1.0	1.0	1.2	1.2
4	0.8	0.8	1.2	1.2
5	0.6	0.6	1.2	1.2
6	0.6	0.4	1.2	1.2

**Table 2 nutrients-14-03852-t002:** Baseline characteristics at cluster and individual level ^1^.

	Intervention A	Intervention B	Control
	Mean (SD) or *n* (%)	Mean (SD) or *n* (%)	Mean (SD) or *n* (%)
*Cluster level*			
Families	25 (28.1)	35 (39.3)	29 (32.6)
Participants	81 (26.2)	127 (41.1)	101 (32.7)
Family size	3.2 (SD 0.8)	3.6 (SD 1.1)	3.5 (SD 1.1)
Parental education			
Minimum secondary school completion ^2^	9 (37.5)	8 (24.2)	8 (28.6)
Bachelors degree or equivalent (3–4 y)	6 (25.0)	12 (36.4)	7 (25.0)
Post-graduate degree (>4 y)	9 (37.5)	13 (39.4)	13 (46.4)
*Individual level*			
Children < 18 y			
*n*	40 (49.4)	64 (50.4)	52 (51.5)
Gender (boys, %)	21 (52.5)	33 (51.6)	27 (51.9)
Age (y)	9.5 (SD 4.2)	9.1 (SD 4.2)	8.4 (SD 3.5)
Age category			
3–7 (y)	17 (42.5)	28 (43.8)	23 (44.2)
8–12 (y)	12 (30.0)	17 (26.6)	20 (38.5)
13–17 (y)	11 (27.5)	19 (29.7)	9 (17.3)
Weight (kg)	40.3 (SD 18.3)	37.3 (SD 19.6)	32.0 (SD 16.1)
Height (cm)	144.1 (SD 25.9)	140.4 (SD 27.4)	133.1 (SD 23.2)
BMI (kg/m^2^)	18.0 (SD 2.9)	17.4(SD 2.8)	16.9 (SD 2.8)
Physical activity			
Sedentary	4 (10.0)	6 (10.0)	6 (12.0)
Moderately active	7 (17.5)	10 (16.7)	6 (12.0)
Vigorously active	29 (72.5)	44 (73.3)	38 (76.0)
Smokers	2 (5.0)	1 (1.7)	0 (0.0)
Adults ≥ 18 y			
*n*	41 (50.6)	63 (49.6)	49 (48.5)
Gender (men, %)	18 (43.9)	29 (46.0)	23 (46.9)
Age (y)	41.5 (SD 9.5)	40.5 (SD 9.0)	40.9 (SD 8.0)
Age category			
18–33 (y)	7 (17.1)	10 (15.9)	9 (18.4)
34–49 (y)	26 (63.4)	42 (66.7)	33 (67.3)
50–65 (y)	8 (19.5)	11 (17.5)	7 (14.3)
Weight (kg)	78.6 (SD 14.3)	77.4 (SD 16.3)	75.5 (SD14.6)
Height (cm)	174.2 (SD 9.7)	174.1 (SD 9.3)	174.0 (SD 8.7)
BMI (kg/m^2^)	25.8 (SD 3.8)	25.6 (SD 5.6)	24.8 (SD 4.1)
Physical activity			
Sedentary	8 (20.0)	8 (13.3)	7 (14.6)
Moderately active	23 (57.5)	30 (50.0)	31 (64.6)
Vigorously active	9 (22.5)	22 (36.7)	10 (20.8)
Smokers	4 (10.0)	5 (8.3)	5 (10.4)

^1^ Values are mean (SD) or *n* (%). Eleven participants had missing data on smoking status, alcohol intake, physical activity, and education. Three participants had missing data on height, and six participants had missing data on weight and BMI. BMI = body mass index, cm = centimeter, kg = kilograms, m = meter, y = year. ^2^ This group also included vocational education, shorter courses, and no further education after secondary school completion.

**Table 3 nutrients-14-03852-t003:** Intake of food groups at baseline and estimated mean change from baseline to follow-up in the three groups (g/10 MJ).

	Intervention A				Intervention B				Control			
	*n* *	Baseline (Median (IQR))	Estimated Mean Change in % (95% Cl)	*p*-Value	*n* *	Baseline (Median (IQR))	Estimated Mean Change in % (95% Cl)	*p*-Value	*n* *	Baseline (Median (IQR))	Estimated Mean Change in % (95% Cl)	*p*-Value
*Food groups*												
Milk and milk products	81/74	392 (220, 607)	−12 (−22, −1)	**0.030**	127/120	314 (197, 514)	−15 (−24, −4)	**0.007**	101/94	385 (218, 517)	−10 (−21, 3)	0.128
Cheese and cheese products	81/70	36 (26, 54)	−35 (−52, −12)	**0.005**	127/120	35 (18, 60)	−17 (−30, −1)	**0.044**	100/94	37 (19, 56)	15 (−5, 39)	0.158
Bread, total	81/74	153 (117, 189)	27 (12, 45)	**0.000**	127/120	160 (129, 197)	22 (12, 32)	**0.000**	101/94	154 (130, 211)	16 (6, 27)	**0.002**
Rye bread	74/70	61 (37, 103)	23 (−17, 82)	0.299	115/114	73 (36, 105)	41 (2, 95)	**0.036**	99/90	72 (39, 105)	−14 (−32, 9)	0.225
Wheat bread	81/74	86 (60, 110)	37 (14, 63)	**0.001**	127/120	87 (55, 119)	27 (9, 48)	**0.002**	101/92	92 (61, 116)	18 (−4, 45)	0.117
Breakfast cereals	60/54	27 (0, 65)	−29 (−57, 19)	0.190	102/95	27 (6, 72)	−10 (−39, 34)	0.602	78/63	30 (6, 67)	−52 (−72, −19)	**0.006**
Rice and pasta	78/71	33 (19, 71)	−14 (−42, 28)	0.456	121/111	42 (27, 69)	−30 (−53, 5)	0.084	97/87	41 (21, 76)	−19 (−42, 13)	0.211
Potatoes and potato products	74/67	31 (16, 45)	43 (−21, 161)	0.239	110/104	33 (13, 62)	26 (−27, 120)	0.409	88/77	23 (10, 45)	35 (−27, 148)	0.336
Vegetables	81/74	169 (119, 209)	9 (−7, 28)	0.292	127/120	194 (128, 292)	−3 (−14, 9)	0.635	101/94	178 (124, 260)	9 (−2, 22)	0.110
Fruit	80/73	114 (49, 172)	46 (9, 95)	**0.009**	125/120	118 (49, 170)	56 (26, 93)	**0.000**	101/94	120 (53, 174)	15 (−6, 40)	0.158
Meat and meat products	81/74	107 (73, 136)	29 (13, 48)	**0.000**	127/119	96 (69, 124)	−6 (−21, 13)	0.507	100/93	98 (66, 130)	15 (1, 30)	**0.034**
Poultry and poultry products	61/47	18 (3, 38)	−51 (−78, 13)	0.092	112/98	23 (9, 44)	−28 (−57, 21)	0.212	81/67	14 (2, 35)	−40 (−66, 6)	0.080
Fish and fish products	61/42	17 (3, 30)	−61 (−79, −28)	**0.003**	107/97	17 (5, 43)	−18 (−47, 27)	0.376	83/67	17 (4, 37)	−52 (−71, −21)	**0.004**
Processed meat, poultry, and fish	81/74	51 (35, 74)	26 (3, 54)	**0.024**	127/119	58 (38, 77)	−3 (−17, 14)	0.709	98/93	51 (36, 85)	4 (−13, 25)	0.664
Butter and spreads	74/69	13 (6, 21)	4 (−26, 48)	0.804	117/111	11 (6, 19)	3 (−21, 34)	0.812	94/87	19 (9, 29)	−2 (−27, 30)	0.878
Cakes, sweets, and chocolate	78/74	83 (53, 119)	38 (−0, 92)	0.053	125/114	90 (60, 134)	−25 (−43,−1)	**0.040**	100/91	89 (64, 113)	−8 (−28, 17)	0.493
Sugar-sweetened beverages	70/64	143 (51, 273)	27 (−32, 136)	0.451	109/108	137 (57, 257)	58 (−11, 183)	0.120	83/80	96 (35, 216)	69 (−12, 222)	0.116
*Bread fillings*												
Fillings, total	81/72	75 (45, 115)	−13 (−34, 14)	0.306	126/118	81 (58, 127)	−17 (−29, −3)	**0.017**	101/94	80 (56, 111)	−15 (−28, 0)	0.051
High-salt (>2.5 g/100 g)	68/62	8 (2, 15)	14 (−31, 88)	0.608	106/97	9 (3, 22)	−24 (−49, 12)	0.163	83/80	9 (2, 18)	13 (−27, 77)	0.579
Medium-salt (1–2.5 g/100 g)	78/66	39 (14, 59)	−25 (−51, 15)	0.186	118/110	33 (17, 53)	−10 (−33, 20)	0.463	96/86	41 (24, 57)	−31 (−50, −5)	**0.024**
Low-salt (<1.0 g/100 g)	76/69	30 (11, 45)	−11 (−41, 34)	0.582	121/110	31 (16, 55)	−24 (−45, 6)	0.114	98/85	26 (15, 45)	−43 (−61, −16)	**0.004**

All analyses are based on imputed datasets and presented as mean change in % with 95% CI calculated using mixed models with age, gender, BMI, parental education, under- and acceptable energy intake as fixed effects and participant as random effect. Baseline values are based on observed values and include participants with zero intake. All values were log-transformed before analysis. *p*-values in bold are significant. * Number of participants with an intake > 0 g/day at baseline/follow-up.

**Table 4 nutrients-14-03852-t004:** Estimated differences between groups at follow-up in intake of food groups (g/10 MJ).

		Intervention A Compared to Control		Intervention B Compared to Control		Intervention B Compared to Intervention A	
	ICC	Mean Difference in % (95% CI)	*p*-Value	Mean Difference in % (95% CI)	*p*-Value	Mean Difference in % (95% CI)	*p*-Value
*Food groups*							
Milk and milk products	0.14	3 (−16, 27)	0.743	−6 (−22, 14)	0.541	−9 (−25, 11)	0.356
Cheese and cheese products	0.22	−38 (−58, −11)	**0.011**	−24 (−45, 7)	0.117	24 (−13, 78)	0.235
Bread, total	0.24	1 (−14, 18)	0.891	6 (−7, 21)	0.396	5 (−10, 22)	0.531
Rye bread	0.06	20 (−23, 88)	0.424	44 (−3, 114)	0.073	20 (−22, 84)	0.409
Wheat bread	0.44	6 (−25, 50)	0.732	8 (−21, 47)	0.623	2 (−27, 41)	0.915
Breakfast cereals	0.30	59 (−32, 270)	0.283	130 (7, 395)	**0.033**	45 (−36, 225)	0.370
Rice and pasta	0.37	10 (−46, 123)	0.791	2 (−46, 95)	0.943	−7 (−53, 83)	0.835
Potatoes and potato products	0.49	43 (−49, 305)	0.497	13 (−56, 190)	0.797	−21 (−71, 113)	0.641
Vegetables	0.34	−3 (−23, 21)	0.769	−9 (−26, 11)	0.341	−6 (−25, 17)	0.571
Fruit	0.18	17 (−16, 62)	0.346	31 (−2, 75)	0.066	12 (−18, 53)	0.470
Meat and meat products	0.46	14 (−16, 56)	0.403	−20 (−40, 7)	0.131	−30 (−48, −5)	**0.020**
Poultry and poultry products	0.44	−14 (−70, 150)	0.786	163 (0, 594)	0.051	205 (10, 745)	**0.032**
Fish and fish products	0.21	−32 (−70, 51)	0.340	83 (−11, 277)	0.102	171 (25, 487)	**0.012**
Processed meat, poultry, and fish	0.35	30 (−6, 81)	0.115	2 (−24, 37)	0.894	−22 (−43, 7)	0.126
Butter and spreads	-	−16 (−43, 24)	0.390	−15 (−40, 19)	0.345	1 (−30, 45)	0.975
Cakes, sweets, and chocolate	0.47	36 (−22, 138)	0.283	−13 (−48, 44)	0.579	−36 (−63, 9)	0.099
Sugar-sweetened beverages	0.21	12 (−56, 183)	0.817	35 (−42, 211)	0.487	21 (−50, 194)	0.679
*Bread fillings*							
Fillings, total	0.11	3 (−25, 41)	0.871	−1 (−25, 31)	0.950	−3 (−28, 30)	0.819
High-salt (>2.5 g/100 g)	0.16	−5 (−50, 81)	0.870	−25 (−58, 33)	0.322	−21 (−57, 46)	0.450
Medium-salt (1–2.5 g/100 g)	0.14	1 (−43, 78)	0.982	15 (−31, 92)	0.581	15 (−33, 97)	0.620
Low-salt (<1.0 g/100 g)	0.20	24 (−34, 133)	0.506	25 (−29, 120)	0.432	1 (−44, 83)	0.972

Analyses are based on imputed datasets and presented as mean difference in % with 95% CI calculated using mixed models with treatment group, age, gender, BMI, parental education, and under- and acceptable reported energy intake as fixed effects and family as random effect. All values were log-transformed before analysis. *p*-values in bold are significant. ICC = intracluster correlation coefficient.

**Table 5 nutrients-14-03852-t005:** Intake of energy and nutrients at baseline and estimated mean change from baseline to follow-up in the three groups (g/10 MJ).

	Intervention A (*n* = 81)			Intervention B (*n* = 127)			Control (*n* = 101)		
	Baseline Mean (SD)	Estimated Mean Change (95% Cl)	*p*-Value	Baseline Mean (SD)	Estimated Mean Change (95% Cl)	*p*-Value	Baseline Mean (SD)	Estimated Mean Change (95% Cl)	*p*-Value
Energy (kJ/day)	8335 (2517)	−406 (−830, 19)	0.061	7695 (2406)	325 (−20, 670)	0.065	7934 (2012)	−394 (−686, −103)	**0.008**
Macronutrients									
Fat (E%)	35.8 (5.1)	1.0 (0.1, 2.0)	**0.037**	35.2 (4.4)	0.0 (−0.9, 0.9)	0.951	36.9 (5.0)	0.3 (−0.7, 1.3)	0.559
Saturated fat (E%)	13.8 (2.5)	0.0 (−0.5, 0.5)	0.903	13.2 (2.4)	−0.8 (−1.4, −0.3)	**0.001**	14.3 (2.9)	−0.5 (−1.1, 0.1)	0.120
Carbohydrates, total (E%)	48.4 (6.0)	−1.0 (−2.2, 0.1)	0.074	49.1 (5.2)	0.1 (−0.9, 1.1)	0.837	48.0 (5.5)	−0.2 (−1.3, 0.9)	0.691
Added sugar (E%)	10.1 (5.7)	−0.2 (−1.2, 0.8)	0.643	8.7 (3.8)	0.8 (0.0, 1.5)	**0.045**	9.1 (4.9)	0.7 (−0.2, 1.6)	0.122
Protein (E%)	15.7 (2.5)	0.0 (−0.6, 0.5)	0.963	15.7 (2.5)	−0.1 (−0.5, 0.3)	0.763	15.1 (2.3)	−0.1 (−0.5, 0.4)	0.800
Dietary fiber (g/10 MJ)	24 (6)	−1 (−2, 0)	0.204	26 (6)	−1 (−2, 0)	0.060	25 (7)	−2 (−3, −1)	**0.001**
Micronutrients									
Sodium (g/10 MJ)	3.8 (0.6)	−0.4 (−0.5, −0.2)	**0.000**	4.0 (0.8)	−0.5 (−0.7, −0.3)	**0.000**	4.0 (0.6)	0.0 (−0.2, 0.2)	0.927
Potassium (g/10 MJ)	3.3 (0.6)	0.1 (−0.0, 0.3)	0.087	3.4 (0.6)	0.1 (−0.1, 0.2)	0.334	3.2 (0.6)	0.0 (−0.1, 0.2)	0.430

All analyses are based on imputed datasets and presented as mean change with 95% CI calculated using mixed models with age, gender, BMI, parental education, and under- and acceptable reported energy intake as fixed effects and participant as random effect. *p*-values in bold are significant.

**Table 6 nutrients-14-03852-t006:** Estimated differences between groups at follow-up in intake of energy and nutrients (g/10 MJ).

		Intervention A Compared to Control		Intervention B Compared to Control		Intervention B Compared to Intervention A	
	ICC	Mean Difference (95% CI)	*p*-Value	Mean Difference (95% CI)	*p*-Value	Mean Difference (95% CI)	*p*-Value
Energy (kJ/d)	0.24	−42 (−513, 429)	0.860	512 (86, 938)	**0.019**	554 (101, 1007)	**0.016**
Macronutrients							
Fat (E%)	0.36	−0.1 (−1.9, 1.7)	0.875	−1.4 (−3.0, 0.3)	0.099	−1.2 (−2.9, 0.5)	0.159
Saturated fat (E%)	0.29	0.1 (−0.9, 1.0)	0.882	−1.0 (−1.8, −0.1)	**0.031**	−1.0 (−2.0, −0.1)	**0.027**
Carbohydrates, total (E%)	0.35	−0.2 (−2.2, 1.8)	0.833	1.2 (−0.6, 3.0)	0.198	1.4 (−0.5, 3.3)	0.153
Added sugar (E%)	0.41	−0.5 (−2.2, 1.2)	0.551	0.0 (−1.5, 1.6)	0.986	0.5 (−1.1, 2.2)	0.524
Protein (E%)	0.32	0.3 (−0.5, 1.1)	0.441	0.2 (−0.6, 0.9)	0.657	−0.1 (−0.9, 0.6)	0.702
Dietary fiber (g/10 MJ)	0.35	0 (−2, 2)	0.679	1 (−1, 3)	0.205	1 (−1, 3)	0.445
Micronutrients							
Sodium (g/10 MJ)	0.57	−0.4 (−0.8, −0.0)	**0.027**	−0.4 (−0.7, −0.0)	**0.026**	0.0 (−0.3, 0.4)	0.829
Potassium (g/10 MJ)	0.30	0.1 (−0.1, 0.3)	0.397	0.0 (−0.2, 0.2)	0.676	−0.1 (−0.3, 0.2)	0.621

Analyses are based on imputed datasets and presented as mean difference with 95% CI calculated using mixed models with treatment group, age, gender, BMI, parental education, and under- and acceptable reported energy intake as fixed effects and family as random effect. *p*-values in bold are significant. ICC = intracluster correlation coefficient.

## Data Availability

The datasets analyzed during the current study are available upon reasonable request from the corresponding author.

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
