# Peer review of "Dietary Effects of Introducing Salt-Reduced Bread with and without Dietary Counselling—A Cluster Randomized Controlled Trial"

_nutrients, 2022, doi:10.3390/nu14183852_

Round 1

Reviewer 1 Report

The present study “Dietary effects of introducing salt-reduced bread with and 2 without dietary counselling – a cluster randomized controlled 3 trial” aims at evaluating the potential benefits of the consumption of salt-reduced bread and dietary counselling on the human diet over a 4-month cluster.

Overall, I believe that the paper is well written and the methods are carefully described. The objectives and the design of the study are clearly stated and identified in the Introduction section. The design and presentation of the result are stated in a well written manner. The paper is well structured and the conclusions are supported by the results. The topic of the present paper is relevant and may be of interest for the readers of the journal.

On the other hand, the introduction should be more in-depth with more references for an accurate overview of the topic in discussion. Tables 3 has way too much data in it and it is not easily readable. The table should be designed in a way that the data presented is easily readable. The conclusions could be more detailed, given the fact that the study has a lot of information in it.   

Reviewer 2 Report

The present cluster RCT study investigating the effect of providing salt-reduced bread both with and without dietary counselling is well conducted and well written. However, I have some concerns:

1.       Study Design and Population: At recruitment, how were the average bread consumption of the families and the exclusion criteria assessed?

2.       Background characteristics: sedentary, moderate and vigorous physical activity should be defined.

3.       row 132: the abbreviation BMI should be written out

4.       The chapter 2.3.5. Under-, acceptable- and over-reported intake of energy is quite hard to read, please clarify the text. BMR?

5.       Results:  In total, 89 families (n=309) were included in the study. 25 families (n=81) were assigned to Intervention A, 35 families (n=127) to Intervention B, 29 families (n=101) to the control group. The numbers in the parentheses are the number of family members (?), please clarify.
Why were there so uneven number of families and participants in the study groups?

6.       What is known about the 5 families which dropped out?

7.       Table 2: the segment All participants could be removed since the sections Adults and Children are provided. Anthropological measures seem so odd when combining children and adults.

8.       How do the authors explain that the intake of ‘bread, total’ increased significantly for all 3 groups, but consumption of ‘butter and spreads’ did not change, and the intake of ‘bread fillings, total’ was significantly reduced in group B and was borderline significantly reduced in the control group?

9.       The intake of ‘cakes, sweets, and chocolate’ and ‘sugar sweetened beverages’ decreased by 25% in the Intervention group B. Nevertheless, the energy intake was increased. This seems odd.

10.   Table 3 is too large for the template and thus hard to read. Hopefully, the editorial office will untangle this problem.
